



# Atmospheric methane source and sink sensitivity analysis using Gaussian process emulation

Angharad C. Stell[1], Luke M. Western[1], and Matthew Rigby[1]

[1]School of Chemistry, University of Bristol, Bristol, BS8 1TS, UK

**Correspondence:** Angharad C. Stell (a.stell@bristol.ac.uk), Matthew Rigby (matt.rigby@bristol.ac.uk)

**Abstract.** We present a method to efficiently approximate the response of atmospheric methane mole fraction and $\delta^{13}$C-CH$_4$ to changes in uncertain emission and loss parameters in a three-dimensional global chemical transport model. Our approach, based on Gaussian process emulation, allows relationships between inputs and outputs in the model to be efficiently explored. The presented emulator successfully reproduces the chemical transport model output with a root-mean-square error of 1.2 ppb

and 0.06 ‰ for hemispheric methane mole fraction and $\delta^{13}$C-CH$_4$, respectively, for 28 uncertain model inputs. The method is shown to outperform multiple linear regression, because it captures non-linear relationships between inputs and outputs, as well as the interaction between model input parameters. The emulator was used to determine how sensitive methane mole fraction and $\delta^{13}$C-CH$_4$ are to the major source and sink components of the atmospheric budget, given current estimates of their uncertainty. We find that our current knowledge of the methane budget, as inferred through hemispheric mole fraction obser-

vations, is limited primarily by uncertainty in the global mean hydroxyl radical concentration and emissions from fresh water. Our work quantitatively determines the added value of measurements of $\delta^{13}$C-CH$_4$, which are sensitive to some uncertain parameters that mole fraction observations on their own are not. However, we demonstrate the critical importance of constraining isotopic initial conditions and isotopic source signatures, small uncertainties in which strongly influence long-term $\delta^{13}$C-CH$_4$ trends, because of the long timescales over which transient perturbations propagate through the atmosphere. Our results also

demonstrate that the magnitude and trend of methane mole fraction and $\delta^{13}$C-CH$_4$ can be strongly influenced by the combined uncertainty of more minor components of the atmospheric budget, which are often fixed and assumed to be well-known in inverse modelling studies (e.g. emissions from termites, hydrates, and oceans). Overall, our work provides an overview of the sensitivity of atmospheric observations to budget uncertainties and outlines a method which could be employed to account for these uncertainties in future inverse modelling systems.

## 1 Introduction

Methane (CH$_4$) is the second most important greenhouse gas in terms of anthropogenic radiative forcing of climate (Myhre et al., 2013; Etminan et al., 2016). It has a wide range of sources and sinks, and the currently estimated magnitude of each





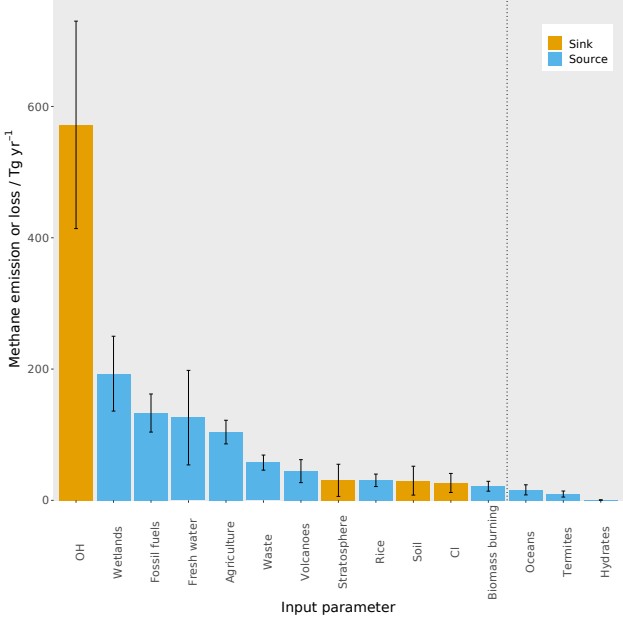

**Figure 1.** The magnitude of the different sources and sinks contributing to the methane budget, according to the combined ranges of bottom-up and top-down estimates (Saunois et al., 2016). The blue bars are sources of methane and the orange bars are sinks of methane. The error bars represent the range of values used in this work, which are the minimum and maximum values given in Saunois et al. (2016). The dashed black line shows the cut-off between the parameters that are varied in this work, and those that are not (see Sect. 2.2 for more detail).

source and sink is shown in Fig. 1. However, the understanding of the atmospheric methane budget is incomplete. This lack of

understanding is demonstrated by a mismatch between bottom-up (inventory and process model-based) and top-down (atmospheric data-based) emissions estimates (Kirschke et al., 2013), and conflicting accounts of the drivers of recent changes in its atmospheric budget; for example, recent studies have proposed that the plateau in methane concentrations in the early 2000s and subsequent growth since 2007 (Rigby et al., 2008), could be driven by increased wetland emissions (Nisbet et al., 2016), increased agricultural emissions (Schaefer et al., 2016), reduced biomass burning and increased fossil fuel sources (Worden

et al., 2017), or (non-statistically significant) changes in hydroxyl radical (OH) concentrations (Rigby et al., 2017; Turner et al., 2017).

Top-down (atmospheric data-based) investigations of the global methane budget have primarily relied on atmospheric measurements of mole fractions made at "background" sites, far from emission sources, (e.g. Houweling et al. (1999); Chen and Prinn (2006); Simpson et al. (2006); Rigby et al. (2008); Bousquet et al. (2011); Turner et al. (2017); Rigby et al. (2017);

Naus et al. (2019)), and/or remotely sensed observations (e.g. Bergamaschi et al. (2013); Turner et al. (2016); Miller et al. (2019)). Measurements of the ratio of methane's most abundant isotopologues, $^{12}CH_4$ and $^{13}CH_4$, have increasingly been used to provide additional constraints on methane's sources and sinks (e.g. Bergamaschi et al. (1998); Quay et al. (1999); Nisbet et al. (2016); Rice et al. (2016); Schaefer et al. (2016); Rigby et al. (2017); Turner et al. (2017); Worden et al. (2017); McNor-





ton et al. (2018)). The two isotopologues are emitted in different ratios from different sources (Whiticar and Schaefer, 2007;

Schwietzke et al., 2016), and are fractionated in the atmosphere by the isotopologues' different rates of loss, with respect to

the sinks (Saueressig et al., 2001). These processes affect the ratio of $^{13}CH_4$ to $^{12}CH_4$ in the atmosphere, often described by

$\delta^{13}C$-$CH_4$ in parts per thousand (‰),

$$\delta^{13}C - CH_4 = \left( \frac{(\frac{^{13}CH_4}{^{12}CH_4})_{\text{sample}}}{(\frac{^{13}CH_4}{^{12}CH_4})_{\text{standard}}} \right) \times 1000, \tag{1}$$

where the standard is Pee Dee Belemnite (Coplen, 2011). This global mean $\delta^{13}C$-$CH_4$ has decreased since the renewed methane

growth in 2007 (Nisbet et al., 2016; Schaefer et al., 2016).

Many studies aiming to identify the cause of observed changes in atmospheric methane have relied on one-dimensional

or two-dimensional (1D or 2D) box models of the atmosphere (e.g. Nisbet et al. (2016); Rigby et al. (2017); Schaefer et al.

(2016); Turner et al. (2017); Worden et al. (2017)). A 2D box model typically splits the atmosphere into a very small number of

latitudinal and vertical boxes, within which zonal mean mole fractions are calculated. These models are known to be limited by

their lack of interannual variation in transport and low spatial resolution. Naus et al. (2019) found that 2D box model parameters

could be derived from a three-dimensional chemical transport model (3D CTM) to combat these limitations, although some

bias remained. Global inversions using 3D CTMs have been carried out (e.g. Bousquet et al. (2011); Bergamaschi et al. (2013);

Rice et al. (2016); McNorton et al. (2018)). However, these studies often rely on assumptions of linearity, Gaussian probability

distributions (which can be non-physical) and frequently omit the exploration of some key parameters (e.g. by assuming fixed

and known OH concentrations).

The problem of efficiently estimating the relationship between uncertain inputs and observable outputs of a complex model

has been addressed in other fields using emulation. An emulator is a statistical approximation to a more complex model, often

using a Gaussian process (O'Hagan, 2006; Ebden, 2015). This technique has been applied to a large variety of scientific prob-

lems, for example: parameter optimisation in models describing galaxy formation (Vernon et al., 2010), influenza epidemics

(Farah et al., 2014), and the Greenland ice sheet (Chang et al., 2014); uncertainty quantification in models of biospheric carbon

flux (Kennedy et al., 2008), aerosol effective radiative forcing (Regayre et al., 2018), and concentrations of cloud condensation

nuclei (Lee et al., 2012); and sensitivity analysis in aerosol models (Lee et al., 2011).

In this work, we outline a set of emulators, which simulate atmospheric methane based on the inputs to a 3D CTM. We limit

our investigation to the simulation of hemispheric monthly average mole fraction and $\delta^{13}C$-$CH_4$, and therefore the emulators

effectively serve as a more accurate 2D box model. However, as discussed in Sect. 2.3, we anticipate that it would be trivial to

extend the technique to the simulation of model outputs at individual monitoring sites, or for remotely sensed observations.

To demonstrate the value of the approach, we use the emulators to carry out a sensitivity analysis of atmospheric observations

to the major uncertain components of the methane budget. One-at-a-time sensitivity tests (where only one input parameter is

perturbed at a time) are often used for complex models, due to the computational burden of the large number of simulations





required to carry out a full sensitivity analysis that allows for the possibility of interacting parameters. For example, this
approach is effectively taken in previous methane inverse modelling studies, where sensitivities of observations to bulk regional
flux changes are calculated using finite differences (Fung et al., 1991; Hein et al., 1997; Chen and Prinn, 2006; McNorton et al.,
2018). A variance-based sensitivity analysis (Saltelli et al., 2000), where sensitivities are calculated using a large ensemble
(typically millions) of simulations, would be prohibitive with the computational burden of a 3D CTM. However, here we show

how a variance-based analysis can be performed using $\sim 10^2$ 3D CTM simulations, requiring only modest computational
resources. Using a fast emulator, we are not only able to thoroughly sample the parameter space, but are also able to quantify
parameter interactions, both of which can be critical for an accurate sensitivity analysis of a complex model (Saltelli and
Annoni, 2010). Such a sensitivity analysis, which as far as we are aware has not yet been carried out for the sensitivity of
atmospheric methane to sources and sinks, will allow a better understanding of complex systems controlling atmospheric

abundance of methane and future prioritisation of research into its most important uncertain parameters.

## 2   Methods

This section begins with the motivation for using emulation for sensitivity analysis (Sect. 2.1). Section 2.2 presents the 3D
chemical transport model (CTM), for which the emulator will act as a surrogate model, and its input parameters. Section 2.3
outlines how the model was used to produce the data required to train the emulator. Then, Sect. 2.4 details the mathematical

background to Gaussian process emulators, and their validation method is outlined in Sect. 2.5. Finally, Sect. 2.6 presents the
sensitivity analysis method.

### 2.1   Approach

In order to make running $\sim 10^6$ simulations for a variance-based sensitivity analysis feasible, emulators that are as computa-
tionally cheap as 2D box models were built. The emulators built in this work are a statistical approximation to the 3D CTM

output of hemispheric median monthly methane mole fraction and $\delta^{13}C$-$CH_4$. These emulators are much less computationally
expensive than the 3D CTM, with a single evaluation taking 40 ms to run on a single core of a 1.6 GHz Intel Core i5 CPU in a
laptop, compared to 4.5 hours on 12 cores of a 2.4 GHz Intel E5-2680 v4 Broadwell CPU in a supercomputer for the 3D CTM.
This computational expense reduction is possible while maintaining the spatial resolution in the emissions, loss fields, and
transport, as well as the interannual variability in transport lost in 2D box models. Additionally, this method does not assume

linear relationships between inputs and outputs nor non-interacting inputs, and allows a thorough exploration of the parameter
space and error quantification that is difficult to achieve for 3D CTMs. Perhaps the greatest drawback of the emulation method
in this work is the small number of parameters than can be included, which is further discussed in Sect. 3.1.

In this work, a Gaussian process, which is a type of non-parametric curve fitting, emulates the 3D CTM (further explained
in Sect. 2.4). There are other methods that could be used to create the emulators if the form of the function that maps model

inputs to outputs is known, for example, linear regression if the underlying function is linear. The Gaussian process achieves





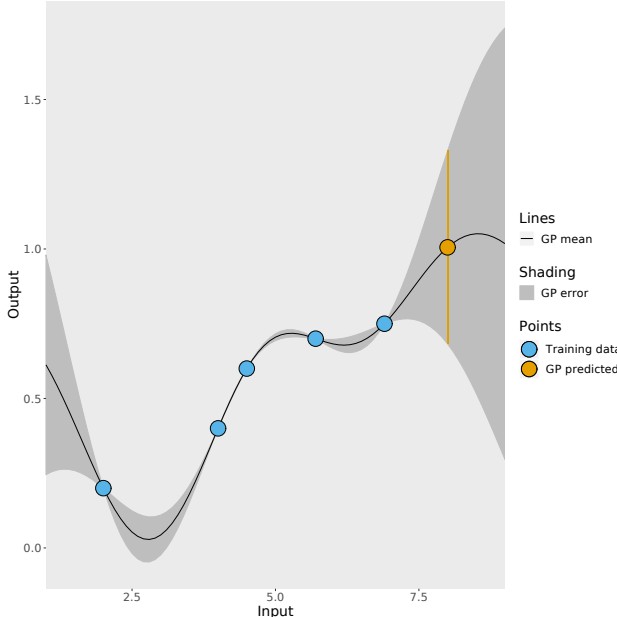

**Figure 2.** A simple 1D example of a Gaussian process (GP). The blue points represent known outputs of the simulator, and the black line is the mean of the Gaussian process which interpolates between the known outputs. The Gaussian process estimated uncertainty in its prediction is represented by the grey shading. The orange point is the Gaussian process prediction of an unknown simulator output and the orange bar represents the uncertainty in the prediction.

the same outcome but does not assume the underlying functional form, and it requires only one main assumption: the outputs follow a multivariate Gaussian distribution. Figure 2 shows a simple example of a 1D Gaussian process emulator. The starting point for a Gaussian process is a set of known simulator outputs (the blue points in Fig. 2), known as a training dataset. As long as a training dataset exists, or can be generated, this emulation method can be applied to any simulator. The Gaussian process

predicts the simulator output at untested inputs by interpolating between the training dataset. The prediction of the simulator output (the black line in Fig. 2) is accompanied by an estimated uncertainty in the prediction (the grey shading) that varies depending on how close the prediction input value is to a value in the training dataset. A prediction of the simulator output (the orange point in Fig. 2) has an uncertainty (shown by the orange bar), which is large if the input value lies beyond the training dataset. Large errors like this are avoided in this work by using a training dataset range that encompasses the full parameter

uncertainty range explored in our sensitivity analysis.

The first step in this method is to decide on the range of possible input parameters to the simulator, and run simulations sampled over these ranges to form a training dataset. A dataset of known model outputs that is independent to the training dataset are used to validate the emulators. Once the emulators are validated, they can be used for the sensitivity analysis.





## 2.2 The chemical transport model setup and input parameter ranges

### 2.2.1 The chemical transport model setup

This section describes how the 3D CTM, which the emulators will approximate, is setup. The model used is the well-established Model for Ozone and Related Chemical Tracers (MOZART) (Emmons et al., 2010), an offline, global 3D CTM. In this work, 56 vertical model levels were used, spanning from the Earth's surface up to about 48 km. The model was run with a spatial resolution of $12.00\,°\,N \times 11.25\,°\,W$, and a time step of one hour, with data output on a 6-hourly basis, using MERRA reanalysis
meteorological fields (Rienecker et al., 2011) from 1995 to 2012.

The MOZART input parameters that are explored in this work describe the methane sources and losses, in a similar way to Ganesan et al. (2018). The sources we use as inputs to MOZART are: wetlands (Bloom et al., 2017), fresh water (see Supplement), agriculture (Crippa et al., 2018), rice (Yan et al., 2009), waste (Crippa et al., 2018), fossil fuels (Crippa et al., 2018), biomass burning (van der Werf et al., 2010), volcanoes (Etiope and Milkov, 2004), termites (Fung et al., 1991), hydrates
(Fung et al., 1991), and oceans (Lambert and Schmidt, 1993; Houweling et al., 1999). The loss processes included in the model are the reactions of methane with the hydroxyl radical (OH) (offline, using fields generated by Spivakovsky et al. (2000)), tropospheric chlorine radicals (Cl) (Sherwen et al., 2016), net stratospheric loss (due to reaction with Cl and $O(^1D)$) (Velders, 1995; Patra et al., 2011), and methanotrophic loss in soils (Murguia-Flores et al., 2018). The model input fields are summarised in Table 1.

The $\delta^{13}C$-$CH_4$ observations are modelled by simulating both $^{12}CH_4$ and $^{13}CH_4$. The emissions of these two species are determined by the literature source signatures (Sect. 2.2.2), and the loss differs between the isotopologues according to the literature kinetic isotope effect (Sect. 2.2.2).

For each model simulation, MOZART was spun up using 30 years of repeating meteorology and sources and sinks (nominally representative of the year 1995), starting from a steady state atmosphere. The model is then run for 1996-2012 with time
varying meteorology, emissions, and losses. To account for any transient signals during the first few years following spin-up (further discussed in Sect. 3.5), only 2000-2012 was analysed. In each simulation, the fields in Table 1 provide the spatial and temporal distribution of the emissions and losses. The total global magnitude of the fields are scaled by the range of values discussed in Sect. 2.2.2 in order to investigate the sensitivity of the methane observations.

### 2.2.2 The chemical transport model input ranges

We test the sensitivity to five properties of input source and sink parameters: their source magnitudes, source $\delta^{13}C$-$CH_4$, loss magnitudes, temporal trend variation for the largest emissions or losses, and initial conditions. Several minor terms in the methane budget (termites, hydrates, oceans, and loss kinetic isotope effects) were held constant, and so are not included as inputs to the emulators, in order to simplify the analysis. The uncertainty that results from these minor terms being held





**Table 1.** The emission and loss fields input to MOZART, their temporal resolution, the years covered by the fields and their literature sources.

| Source | Reference | Temporal resolution | Years |
|---|---|---|---|
| Wetlands | Wetcharts (Bloom et al., 2017) | monthly | 2001-2012 |
| | | | (1996 – 2000 are 2001 repeating) |
| Fresh water | This work (see Supplement and available to download (Stell, 2020a)) | annual | climatology |
| Agriculture | EDGAR 4.32 (Crippa et al., 2018) | annual | 1996-2012 |
| Rice | Yan et al. (2009) | monthly | 2000 repeating |
| Waste | EDGAR 4.32 (Crippa et al., 2018) | annual | 1996-2012 |
| Fossil fuel (includes biofuel) | EDGAR 4.32 (Crippa et al., 2018) | annual | 1996-2012 |
| Biomass burning | GFED4s (van der Werf et al., 2010) | monthly | 1997-2012 |
| | | | (1996 is the mean of all years) |
| Volcanoes | Etiope and Milkov (2004) | annual | climatology |
| Termites | Fung et al. (1991) | annual | climatology |
| Hydrates | Fung et al. (1991) | annual | climatology |
| Oceans | Lambert and Schmidt (1993); Houweling et al. (1999) | annual | climatology |
| Loss | | | |
| OH | Spivakovsky et al. (2000) | monthly | climatology |
| Stratosphere | Velders (1995); Patra et al. (2011) | monthly | climatology |
| Cl | Sherwen et al. (2016) | monthly | 2005 repeating |
| Soil | Murguia-Flores et al. (2018) | monthly | 1996-2009 |
| | | | (2010-2012 is 2009 repeating) |

Rice is considered separately to agriculture and wetlands. Biofuel is included in fossil fuel rather than biomass burning. Agricultural burning is included in biomass burning rather than agriculture. The mean of the WetCharts ensemble is used for wetland emissions.

constant is explored in Sect. 2.5. The range of possible values for the chosen parameters must be identified so that a set of

MOZART simulations over these ranges can be created, which forms the training dataset for the emulators.

The ranges of possible source magnitudes were based on the ranges of compiled literature values in Saunois et al. (2016), and the ranges of possible $\delta^{13}$C-CH$_4$ source signatures were the three standard deviation ranges in Schwietzke et al. (2016). The ranges of source parameter values used in this work are given in Table 2.

The ranges of possible loss magnitudes were taken from Saunois et al. (2016), and the kinetic isotope effects were held

constant at typical literature values (King et al., 1989; Tyler et al., 1994; Saueressig et al., 1995; Reeburgh et al., 1997; Crowley





et al., 1999; Snover and Quay, 2000; Tyler et al., 2000; Saueressig et al., 2001) derived as outlined in the Supplement. The reaction rates of methane with OH, Cl, and O($^1$D) were held constant at the values in Burkholder et al. (2015). While there is some uncertainty in these rate constants, the sensitivity to this term will be similar to that of their respective loss magnitudes. The ranges of loss parameter values used in this work are given in Table 2.

The default temporal trends of the emissions and losses from 1996 to 2012 are set by the input fields in Table 1. The overall inventory or process model trend for the five largest methane emissions or losses (OH, wetlands, fresh water, agriculture, and fossil fuels) was allowed to vary by a linear trend of $\pm 20$ %. For example, a trend parameter that reduces a term by 20 % is applied as a 10 % increase in the first year, decreasing to no change in the middle of the time series, and then decreasing to -10 % in the final year.

Three parameters were varied during the spin-up: the total source magnitude, the total source $\delta^{13}$C-CH$_4$, and an overall imbalance between the source and sink. This setup was used to allow three degrees of freedom in the initial mole fraction and $\delta^{13}$C-CH$_4$ field. Table 2 gives the range of these spin-up parameters.

## 2.3  Creating the chemical transport model training and validation datasets

This section discusses the generation of the training and validation datasets, which is the most computationally expensive
part of the analysis, as repeated runs of the 3D CTM are required. The training and validation datasets were designed to give accurate emulators for the whole range of the parameter values in Table 2. Therefore, the sets of input parameters in the datasets should be evenly spaced, so that every possible input parameter set is close to training data. Hence, each parameter described in Table 2 is assigned a uniform probability distribution over the range given. In order to sample from the distributions in a way that effectively covers the input parameter space, a maximin Latin hypercube was used (McKay et al., 1979; Morris and
Mitchell, 1995). A training dataset of 270 MOZART simulations was created and used to build the Gaussian process emulators. An independent maximin Latin hypercube design of 90 MOZART simulations was created as a validation dataset, which was used to evaluate the emulators.

      Although observations were not required for this study, for consistency with observed trends, we opted to calculate hemi-spheric averages based on mole fractions and $\delta^{13}$C-CH$_4$ at grid cells where baseline observations were made by the Global
Monitoring Laboratory (GML) Carbon Cycle group (part of National Oceanic and Atmospheric Administration (NOAA) (Dlu-gokencky et al., 1994, 2017)) and the Institute of Arctic and Alpine Research (INSTAAR) (Miller et al., 2002; White et al., 2018), respectively. Measurement stations that do not have approximately continuous records for the period of interest (more than 9 out of 13 years) were discarded. We also discarded measurement sites that exhibited substantial above-baseline variabil-ity in the model (likely an artefact of the coarse model resolution).

The MOZART outputs are monthly time series describing the southern hemisphere mole fraction, the northern hemisphere mole fraction, the southern hemisphere $\delta^{13}$C-CH$_4$, and the northern hemisphere $\delta^{13}$C-CH$_4$. These four 3D CTM outputs are





**Table 2.** A table of the ranges of the 28 input parameters to MOZART that were varied in the training simulations, hence also in the emulators, and in the sensitivity analysis. Where one value is given, the value is held constant for all simulations. Where two values are given, they are the lower and upper limit, respectively.

| Source | Magnitude / Tg yr$^{-1}$ | Delta value / ‰ | Trend / % |
|---|---|---|---|
| Wetlands | 136, 250 | -63.3, -59.7 | -20, 20 |
| Fresh water | 54, 198 | -64.6, -59.8 | -20, 20 |
| Agriculture | 86, 122 | -75.2, -58.4 | -20, 20 |
| Rice | 21, 40 | -66.0, -58.2 | |
| Waste | 46, 69 | -57.7, -53.5 | |
| Fossil fuel (includes biofuel) | 104, 162 | -45.1, -38.4 | -20, 20 |
| Biomass burning | 14, 29 | -27.9, -16.5 | |
| Volcanoes | 27, 62 | -46.1, -41.9 | |
| Termites | 9.6 | -65.0 | |
| Hydrates | 0 | -62.2 | |
| Oceans | 16 | -57.9 | |
| **Loss** | **Magnitude / Tg yr$^{-1}$** | **Kinetic isotope effect** | **Trend / %** |
| OH | 414, 730 | 1.0039 | -20, 20 |
| Stratosphere | 6, 55 | 1.0397 | |
| Cl | 12, 41 | 1.0640 | |
| Soil | 8, 52 | 1.0215 | |
| **Spin-up** | **Magnitude / Tg yr$^{-1}$** | **Delta value / ‰** | |
| Spin-up source | 495, 976 | -59.5, -52.4 | |
| Spin-up source minus loss | 10, 65 | | |

the quantities that the Gaussian processes emulate. However, it should be trivial to extend this to individual grid cells of the 3D CTM in future work. This number of emulators is feasible as the same training dataset could be used, and the computational burden of both building and running the emulator is far smaller than creating the 3D CTM training simulations.

In order to explore sensitivities to quantities that are more often used (either implicitly or explicitly) to inform the global methane budget, the hemispheric outputs are combined as a global mean, inter-hemispheric difference, and trend of the mole fraction and $\delta^{13}$C-CH$_4$. The global mean is defined as the temporal mean of the mean over the northern and southern hemispheres for all months between 2000 and 2012. The inter-hemispheric difference is the temporal mean over the northern hemisphere minus the southern hemisphere, averaged over all months between 2000 and 2012. The trend is defined as the

global mean in December 2012 minus December 2000.





## 2.4 Gaussian process emulators

### 2.4.1 The basics of Gaussian process emulation

The Gaussian process is defined by two functions that vary depending on the input parameter values: the mean function and the covariance function. It is sufficient to have a mean function of zero, though in this work, a multiple linear regression was chosen

as the system is close to linear. The covariance function is a measure of the similarity of input sets, and as the distance between the inputs increase, the value of the function decreases. In this work we use the squared exponential covariance function as there are no discontinuities or sharp changes in the methane observations due to input parameter variation. The $(i, j)^{\text{th}}$ element of the covariance matrix $(K)$ is given by

$$\eta_{ij} = \sigma_f^2 \exp\left(-\sum_{k=1}^{m} \frac{(x_{k,i} - x_{k,j})^2}{l_k^2}\right), \tag{2}$$

where the maximum covariance is $\sigma_f^2$, $x_k$ and $x_k{}'$ are the values of the $\text{k}^{\text{th}}$ input parameter, and $l_k$ is the length scale parameter to be optimised during training.

The prediction of an output value $(\boldsymbol{y}_*)$ at a set of input parameters $(\boldsymbol{x}_*)$ samples from the joint multivariate Gaussian distribution of the training data $(\boldsymbol{y})$ and the predicted values, which follows

$$\begin{bmatrix} \boldsymbol{y} \\ \boldsymbol{y}* \end{bmatrix} \sim \mathcal{N}\left(m(\boldsymbol{x}*), \begin{bmatrix} K(\boldsymbol{x}, \boldsymbol{x}) & K(\boldsymbol{x}, \boldsymbol{x}*) \\ K(\boldsymbol{x}*, \boldsymbol{x}) & K(\boldsymbol{x}*, \boldsymbol{x}*) \end{bmatrix}\right), \tag{3}$$

where $m$ is the mean function and $\boldsymbol{x}$ is the training dataset inputs. This means that the expected value of $\boldsymbol{y}*$ is

$$E(\boldsymbol{y}*) = m(\boldsymbol{x}*) + K(\boldsymbol{x}*, \boldsymbol{x})K(\boldsymbol{x}, \boldsymbol{x})^{-1}\boldsymbol{y}, \tag{4}$$

and the uncertainty, in terms of variance, in the estimate is

$$V(\boldsymbol{y}*) = K(\boldsymbol{x}*, \boldsymbol{x}*) - K(\boldsymbol{x}*, \boldsymbol{x})K(\boldsymbol{x}, \boldsymbol{x})^{-1}K(\boldsymbol{x}, \boldsymbol{x}*). \tag{5}$$

The Gaussian process emulation method is further described in Rasmussen and Williams (2006), and some simple tutorials are

available in O'Hagan (2006) and Ebden (2015).

### 2.4.2 Gaussian process emulation for time series outputs

Each MOZART output is a time series of 156 months (12 months for each of 13 years) of hemispheric median mole fraction or $\delta^{13}\text{C-CH}_4$. These 156 monthly outputs are highly correlated in time, which can be exploited in the design of the emulator covariance matrix to minimise information loss. There will also be correlations in space between the northern and southern





hemispheric outputs, but these correlations are not considered in this work. The chosen covariance matrix ($\boldsymbol{\Sigma}$) is composed of the Kronecker product of a temporal covariance matrix ($\boldsymbol{\Sigma}_t$) and a parameter covariance matrix ($\boldsymbol{\Sigma}_x$),

$$\boldsymbol{\Sigma} = \boldsymbol{\Sigma}_t \otimes \boldsymbol{\Sigma}_x. \tag{6}$$

The elements of $\boldsymbol{\Sigma}_t$ and $\boldsymbol{\Sigma}_x$ are described by $\zeta_{ij}$ and $\eta_{ij}$, respectively. The chosen temporal covariance is a first order autoregressive model (its value depends only on the previous month), and its $(i,j)^{\text{th}}$ element is

$$\zeta_{ij} = \frac{\rho^{|t_i - t_j|}}{1 - \rho^2}, \tag{7}$$

where $\rho$ is the autocorrelation parameter and $t$ is the month. The chosen parameter covariance is a squared exponential, and its $(i,j)^{\text{th}}$ element is given by Eq. 2.

The emulator parameters ($\rho$ in Eq. 7, $\sigma_f$ and $l_k$ in Eq. 2) are optimised by maximising the log-likelihood function

$$\log(L) \propto -\frac{1}{2}(\boldsymbol{y} - m(\boldsymbol{x}))^{\text{T}} \boldsymbol{\Sigma}^{-1}(\boldsymbol{y} - m(\boldsymbol{x})) - \frac{1}{2}\log(|\boldsymbol{\Sigma}|). \tag{8}$$

This log-likelihood function is maximised using a bounds constrained quasi-Newton method (Gay, 1990) started from 28 different random points, and the emulator with the maximum log-likelihood is chosen. This setup uses an adaptation of the R package, Stilt (Olson et al., 2018).

### 2.5   Validation of the emulators

It is important to check that the emulators are an accurate approximation of the 3D CTM before they are used. The validation
dataset is used to confirm this, because it contains inputs and known 3D CTM outputs that the emulator was not trained on. The emulator predictions for the validation dataset inputs can be compared to the 3D CTM output, and these differences reveal how accurate the approximation is. There are several graphical comparison methods presented in the Supplement, but the main focus is the absolute error in emulation. For the emulators to be useful, their error in emulating the CTM output must be much smaller than a reasonable estimate of the other errors in the system.

The error in a complex model is difficult to calculate, and so is often ignored, expert judgement is used, or estimates of model-data mismatch uncertainties are approximated (e.g. based on spatial or temporal variability in the model output in the vicinity of observation points, e.g. Chen and Prinn (2006)). In this work, the uncertainty in the 3D CTM is approximated by the uncertainty due to the invariant parameters (as in Vernon et al. (2010)). The invariant parameters and their investigated ranges are given in Table 3. The uncertainty was calculated with a maximin Latin hypercube design of 90 MOZART simulations,
where variations were allowed only in those parameters held constant in the emulator training dataset. This invariant parameter error does not include many other sources of error (e.g. model transport uncertainties are not addressed), and higher-order "invariant parameter errors" (e.g. erroneous trends or spatial distributions), so can be considered a lower bound of the total 3D CTM error.





**Table 3.** The ranges of the invariant parameters explored (from the literature as in Sect. 2.2.2), where the first number is the minimum and the second number is the maximum. The $^{13}CH_4$ A factor is the Arrhenuis pre-exponential factor, which is changed in the model to describe uncertainty in the kinetic isotope effect with respect to the losses. The OH and $^{13}CH_4$ A factor was also considered, but MOZART only allows the rate constant to be input with two decimal places, and the OH and $^{13}CH_4$ A factor is constant when given to two decimal places over the range of kinetic isotope effects explored.

| Term | Magnitude / Tg yr$^{-1}$ | Delta value / ‰ | $^{13}CH_4$ A factor |
|---|---|---|---|
| Termites | 5.0, 14.2 | -66.7, -63.3 | |
| Hydrates | 0.0, 0.9 | -63.0, -61.4 | |
| Oceans | 8.3, 23.7 | -51.7, -44.1 | |
| Soil | | -24.0, -19.0 | |
| Tropospheric chlorine | | | $6.66, 6.68 \times 10^{-12}$ cm$^3$ molecule$^{-1}$ s$^{-1}$ |
| Stratosphere | | | 0.958, 0.966 s$^{-1}$ |

Methane loss by soil was input to the model as negative emissions, hence its isotopic fractionation is not characterised by an A factor.

## 2.6 Calculation of sensitivity indices

The sensitivity analysis, using the validated emulators, identifies how sensitive the 3D CTM outputs are to changes in the inputs. A variance-based sensitivity analysis requires $\sim 10^6$ simulations, which would be unfeasible using the 3D CTM as the model is so computationally expensive. By using an emulator, the only 3D CTM simulations required are those needed to train the emulators.

In a variance-based sensitivity analysis, the model sensitivity is quantified using sensitivity indices. These indices measure
the proportion of the output variance caused by an input parameter being varied over its possible range (Saltelli et al., 2000). In this work, two sensitivity indices are calculated: the first order and total effect indices. The first order sensitivity index reflects the proportion of the variance in the output that can be attributed to a single parameter. This can be calculated as

$$S_k = \frac{V[E(y|x_k)]}{V(y)}, \tag{9}$$

where $V[E(y \mid x_k)]$ is the variance in the expected value of the emulator output $y$ given the value of parameter $x_k$, and $V(y)$
is the variance in the emulator output caused by all parameters.

The total effect index is the proportion of the output variance that can be explained by a single parameter and its interactions with other parameters. This can be calculated as

$$S_{T_k} = 1 - \frac{V[E(y|x_{\sim k})]}{V(y)}, \tag{10}$$

where $V[E(y|x_{\sim k})]$ is the variance in $y$ caused by all parameters except $x_k$. A parameter's interactions with all other parameters
can be calculated by subtracting the first order sensitivity index from the total sensitivity index. These sensitivity indices were calculated using Monte-Carlo methods (Saltelli et al., 2000), and further details are given in the Supplement.





## 3   Results and discussion

Here, we demonstrate the accuracy of the emulators and show how they can be applied to a sensitivity study of the global methane budget. Section 3.1 compares the 3D chemical transport model (CTM) training dataset to the observations, in order
to check that the observations lie within the envelope of the model output ensemble. Section 3.2 examines the size of the 3D CTM invariant parameter error, which is compared to the emulator error in Sect. 3.3 in order to justify the use of emulation. The Gaussian process emulation method is then shown to be warranted by comparison to a simpler multiple linear regression in Sect. 3.4. Having demonstrated the utility of the method, a sensitivity analysis is presented in Sect. 3.5.

### 3.1   Comparison of 3D chemical transport model training dataset to observations

The training dataset is compared to observations to check that the observations lie within the envelope of the MOZART output ensemble. The MOZART simulations used to train the emulators are shown in Fig. 3. The outputs that are considered in the sensitivity analysis (the temporal mean of the global mean, the temporal mean of the inter-hemispheric difference, and the trend in the global mean (Sect. 2.3) for the mole fraction and $\delta^{13}$C-CH$_4$) are presented in Fig. 4. In these figures, the distribution of the MOZART simulations (in orange) is compared to the NOAA and INSTAAR atmospheric observations presented in Rigby
et al. (2017) (in black) (derived from a slightly different subset of measurement stations to those used in this work).

These figures demonstrate the large range of methane mole fraction and $\delta^{13}$C-CH$_4$ values covered by the training dataset. This is caused by the large range of emission and loss values considered, and also the somewhat arbitrary initial condition range. Additionally, the figures show that the observations are within the MOZART range for all outputs.

These figures also show that the range of MOZART inter-hemispheric difference values is small compared to the range
of global mean and trend values. Ideally, the spatial distributions of the emissions and losses would also be parameterised, allowing greater variation in the inter-hemispheric differences. However, only a limited number of parameters can be included in the Gaussian process emulation method of this work. The more parameters, the more 3D CTM simulations are required to train the emulator and the slower computation becomes. Therefore, only up to about 30 parameters are typically included in a Gaussian process, whereas methods such as adjoint models (e.g. Bousquet et al. (2011); Bergamaschi et al. (2013)) can include
thousands of parameters.

### 3.2   The 3D chemical transport model invariant parameter error

The MOZART invariant parameter error (Sect. 2.5), as far as we are aware, has not been considered in previous methane modelling studies. This error was calculated as the standard deviation in the output of the set of simulations where parameters





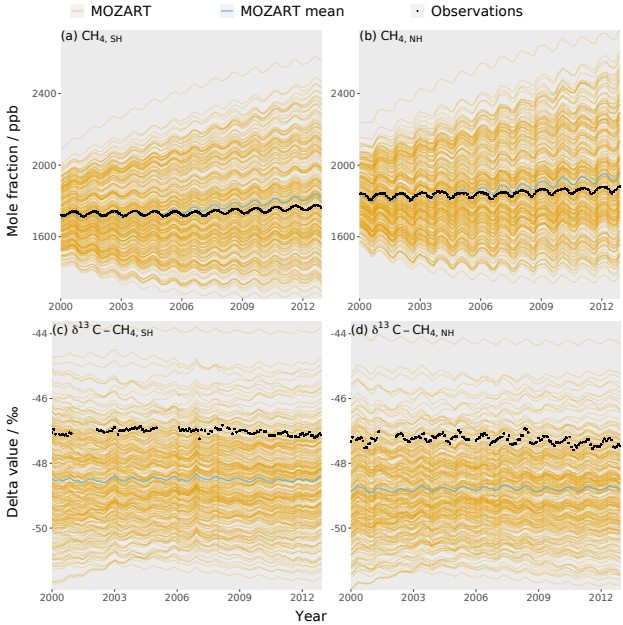

**Figure 3.** The MOZART training dataset (orange lines), the mean MOZART output (blue line), and the observations (black line) for each of the four emulators: (a) the southern hemisphere mole fraction, (b) the northern hemisphere mole fraction, (c) the southern hemisphere $\delta^{13}$C-CH$_4$, and (d) the northern hemisphere $\delta^{13}$C-CH$_4$. The observations are hemispheric averages based on NOAA and INSTAAR data (derived from a slightly different subset of measurement stations to those used in this work) presented in Rigby et al. (2017).

not included in the emulator training dataset (fluxes from termites, hydrates and oceans, as well as isotopic fractionation by soil, tropospheric Cl, and stratospheric losses) were perturbed within their uncertainty ranges (Table 3). Over the 13 year period of our study, the mean invariant parameter uncertainty is about 10 ppb and 0.1 ‰ for the mole fraction and $\delta^{13}$C-CH$_4$, respectively. These values are large compared to atmospheric methane trends (e.g. between 2000 and 2012, the methane mole fraction and $\delta^{13}$C-CH$_4$ changed by around 40 ppb and -0.1 ‰, respectively). Furthermore, these uncertainties are highly correlated through the study period, and therefore effectively act as substantial bias. The omission of this substantial source of error will likely be leading to an underestimation of uncertainties of emissions and losses derived in inverse modelling studies, or may contribute to the misallocation of some emission or loss to particular processes.

### 3.3 Validation of the emulators

Before using the emulators, it is important to check that they reproduce the 3D CTM output well. A more complete analysis can be found in the Supplement, which shows that the emulator is an unbiased representation of the 3D CTM. The emulator error was calculated by predicting the validation dataset (Sect. 2.3) and comparing the predictions to the MOZART output,





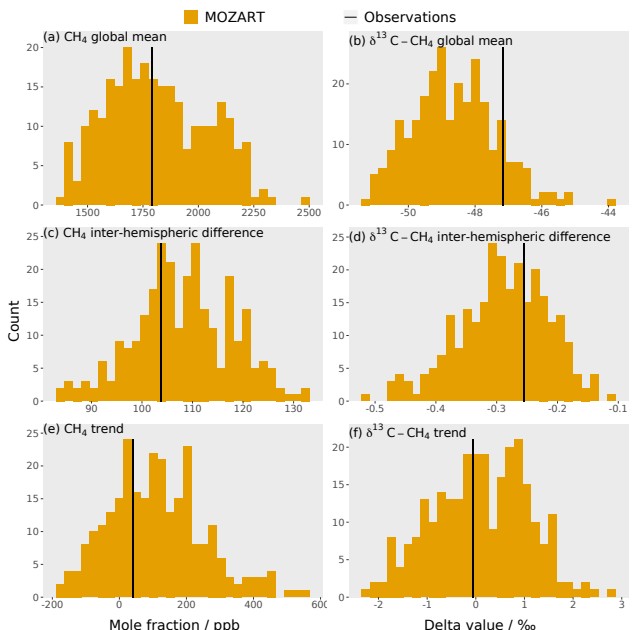

**Figure 4.** Histograms of the 270 3D CTM training simulations for six outputs: (a) mole fraction global mean, (b) $\delta^{13}$C-CH$_4$ global mean, (c) mole fraction inter-hemispheric difference, (d) $\delta^{13}$C-CH$_4$ inter-hemispheric difference, (e) mole fraction trend, and (f) $\delta^{13}$C-CH$_4$ trend. The black line is the corresponding value for the NOAA and INSTAAR atmospheric observations (Sect. 2.3), which are hemispheric averages (derived from a slightly different subset of measurement stations to those used in this work) presented in Rigby et al. (2017).

using the root-mean-square error (RMSE),

$$\text{RMSE} = \sqrt{\sum_{i=1}^{n} \frac{(\boldsymbol{y}_{em,i} - \boldsymbol{y}_{mzt,i})^2}{n}}, \tag{11}$$

where $\boldsymbol{y}_{em}$ is the emulator output, $\boldsymbol{y}_{mzt}$ is the MOZART output, and $n$ is the number of simulations being compared. The RMSE was calculated to be about 1.2 ppb and 0.06 ‰ for the mole fraction and $\delta^{13}$C-CH$_4$, respectively. This emulator error is small when compared to the MOZART invariant parameter error (Sect. 2.5) in Fig. 5.

As the MOZART invariant parameter error is significantly larger than the emulator error, it is possible to use a less accurate emulator that requires fewer training simulations. As making the training dataset is the longest step in the process, this would be beneficial for more time-consuming higher resolution models. In the case of MOZART, we find that only around 90 simulations may be required, which is further discussed in the Supplement.

## 3.4 Comparison of multiple linear regression and the Gaussian process

Previous studies (e.g. McNorton et al. (2018)) have assumed that for small changes in the source and loss magnitudes, the relationship between methane sources and losses and atmospheric mole fraction and $\delta^{13}$C-CH$_4$ is linear and that the parameters



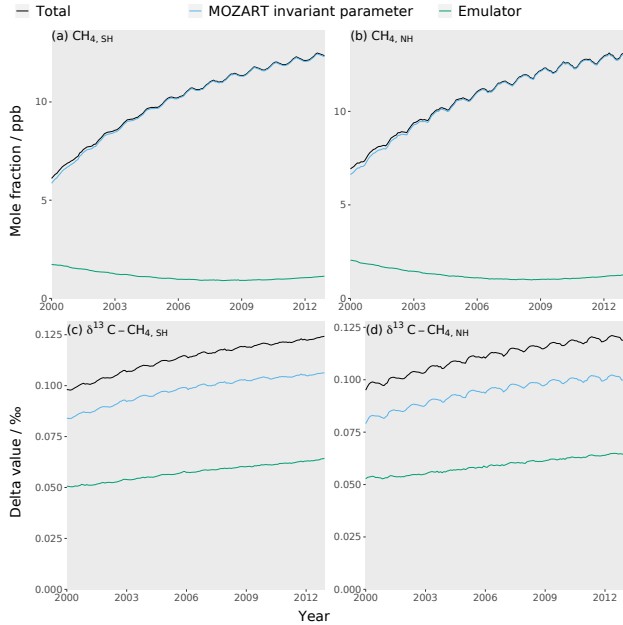

**Figure 5.** The MOZART error (blue line), emulator error (green line), and total error (MOZART and emulator errors added in quadrature) (black line) for each of the four emulators: (a) the southern hemisphere mole fraction, (b) the northern hemisphere mole fraction, (c) the southern hemisphere $\delta^{13}$C-CH$_4$, and (d) the northern hemisphere $\delta^{13}$C-CH$_4$.

do not interact (Sect. 3.5). If these two conditions are true, or close to true, then multiple linear regression would be able to emulate the 3D CTM. Multiple linear regression might be preferred to a Gaussian process as it requires a smaller training

dataset (hence fewer 3D CTM simulations) and is conceptually and computationally simpler. Therefore, this section compares the performance of multiple linear regression and the Gaussian process as emulators of the 3D CTM.

The residuals for the global mean between the 3D CTM validation dataset and the predictions from the two methods (multiple linear regression and the Gaussian process) are compared in Fig. 6. The Gaussian process residuals, with a RMSE of 1.0 ppb and 0.06 ‰, are much smaller than for multiple linear regression, which are 18 ppb and 0.17 ‰. In comparison to the MOZART

invariant parameter error (10 ppb and 0.1 ‰), the multiple linear regression residuals are large, unlike the Gaussian process (Sect. 3.3). Therefore, the multiple linear regression struggles to emulate MOZART with the required accuracy.

The multiple linear regression accuracy can be improved by considering the non-linearity of the mole fraction with respect to the OH loss. By using a log-transformed OH parameter to estimate the mole fraction, the RMSE becomes 11 ppb (the complete residual distribution is shown in Fig. 6). Multiple linear regression using a log-transformed OH parameter still has a signifi-

cantly larger RMSE than the Gaussian process, implying that the remaining small non-linearities and parameter interactions are important for predicting the output value. This finding suggests that inverse modelling studies that have assumed linear



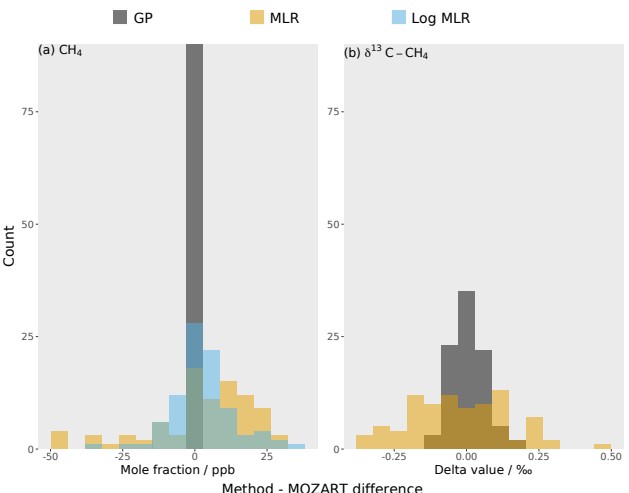

**Figure 6.** The residuals for the global mean between the different emulation methods (in different colours) and the true MOZART output for (a) methane mole fraction and (b) $\delta^{13}$C-CH$_4$. Each emulator is built using a Gaussian process (GP) (grey) or multiple linear regression (MLR) (orange). The mole fraction has an additional emulator: a multiple linear regression with log transformed OH (blue).

and independent sensitivities between observations and source and sink parameters may have under-estimated their posterior uncertainties.

## 3.5 Using the emulators for sensitivity analysis

### 3.5.1 First order sensitivity indices

In this section, we examine the sensitivity of the MOZART outputs to the input parameters describing methane sources and sinks. This sensitivity is explored using the first order sensitivity indices (Eq. 9) in Fig. 7, which show the proportion of the variance of the MOZART output caused by varying each parameter.

The sensitivity of the global mean mole fraction is shown in Fig. 7a, and is dominated by the OH loss magnitude (73 %), with considerable contributions from the freshwater (13 %) and wetlands (8 %) source magnitudes. These sensitivities follow the absolute size of the uncertainty in the source and loss magnitudes seen in Fig. 1, and are therefore relatively unsurprising. However, these results highlight the overwhelming importance of global mean OH concentration in determining the global methane mole fraction, and the major influence of freshwater emission uncertainties, which have largely been ignored in recent global modelling studies.

Figure 7b shows the sensitivity of the global mean $\delta^{13}$C-CH$_4$ to each input parameter. The parameters that this output is most sensitive to are: the Cl sink magnitude (27 %), the agricultural source $\delta^{13}$C-CH$_4$ (16 %), and the initial condition source $\delta^{13}$C-CH$_4$ (16 %), with several other parameters contributing substantially: the freshwater source magnitude (10 %), the stratospheric





loss magnitude (6 %), and the wetlands source magnitude (5 %). As the mole fraction and $\delta^{13}$C-CH$_4$ are most sensitive to different parameters, this means that the $\delta^{13}$C-CH$_4$ could be a useful additional measurement for constraining the methane

budget. However, two of the parameters that $\delta^{13}$C-CH$_4$ is most sensitive to are $\delta^{13}$C-CH$_4$-specific (the agricultural source $\delta^{13}$C-CH$_4$ and the initial condition source $\delta^{13}$C-CH$_4$), and so do not, on their own, add information about the magnitudes of the different methane sources and sinks. Unlike the global mean mole fraction, the ordering of the parameters to which $\delta^{13}$C-CH$_4$ is most sensitive does not simply follow the absolute magnitude of uncertainty in the input parameters. The global mean $\delta^{13}$C-CH$_4$ is most sensitive to the Cl loss magnitude, which has a small uncertainty in comparison to other parameters.

However, this loss is highly fractionating, so it has a large impact on the $\delta^{13}$C-CH$_4$. The second highest contribution to the output variance is the agricultural source $\delta^{13}$C-CH$_4$, which has a large uncertainty compared to other source $\delta^{13}$C-CH$_4$ signatures. Additionally, this source $\delta^{13}$C-CH$_4$ signature is significantly more negative than the atmospheric $\delta^{13}$C-CH$_4$ in comparison to other sources, and so this parameter results in a large output variance in the global mean $\delta^{13}$C-CH$_4$. The global mean $\delta^{13}$C-CH$_4$ is also highly sensitive to the initial conditions due to the long response time of $\delta^{13}$C-CH$_4$ in the atmosphere

compared to the 17 years examined in this work (Tans, 1997).

The mole fraction inter-hemispheric difference (the temporal mean over the northern hemisphere minus the southern hemisphere as in Sect. 2.3) is most sensitive to the freshwater (66 %), fossil fuel (15 %), and wetlands (8 %) source magnitudes, as shown in Fig. 7c. The sensitivity to these parameters is due to their large uncertainties and large differences in emissions between the two hemispheres. The OH loss magnitude, which has the largest uncertainty of any parameter, has been assumed

to be close to equally distributed between the hemispheres (Patra et al., 2014), hence its low sensitivity with respect to this output. The dominant role of freshwater emission uncertainty in determining the inter-hemispheric difference further highlights the need to better understand this part of the methane budget.

Figure 7d shows that the sensitivity of $\delta^{13}$C-CH$_4$ inter-hemispheric difference. The parameters that the $\delta^{13}$C-CH$_4$ inter-hemispheric difference is most sensitive to are: the initial condition source $\delta^{13}$C-CH$_4$ (22 %), the Cl sink magnitude (18 %),

and the fossil fuel source $\delta^{13}$C-CH$_4$ (12 %). There are also significant contributions from the stratospheric loss magnitude (11 %), the OH loss magnitude (9 %), and the wetlands source magnitude (5 %). The parameters to which the $\delta^{13}$C-CH$_4$ inter-hemispheric difference is most sensitive are similar to those that most strongly influence the global mean $\delta^{13}$C-CH$_4$, but with a higher sensitivity to parameters with a large inter-hemispheric difference (e.g. fossil fuels).

The trends (the global mean in December 2012 minus December 2000 as in Sect. 2.3) for the mole fraction and $\delta^{13}$C-CH$_4$

are shown in Fig. 7e and Fig. 7f, respectively. The trend sensitivities are each dominated by single parameters: 58 % of the variance in the mole fraction trend is caused by the uncertainty in the OH loss magnitude, and 71 % of the $\delta^{13}$C-CH$_4$ variance due to variations in the initial conditions. The OH loss trend (15 %), freshwater source magnitude (9 %), and wetlands source magnitude (6 %) contribute significantly to the mole fraction trend, and the agricultural source $\delta^{13}$C-CH$_4$ (11 %) to the $\delta^{13}$C-CH$_4$ trend. The OH loss parameter's importance for the output mole fraction value stems from the large uncertainty in the OH

loss. The $\delta^{13}$C-CH$_4$ trend is highly sensitive to the initial conditions because of the slow response time in the atmospheric $\delta^{13}$C-





$CH_4$, meaning that the trend is strongly dependent on its initial value (Tans, 1997). A wide range of $\delta^{13}$C-$CH_4$ initial condition values (Table 2) are examined in this work, however the importance of the initial conditions applies to even small ranges. For example, if the $\delta^{13}$C-$CH_4$ initial condition is perturbed by 0.1 ‰ from the initial median parameter values, the output atmospheric $\delta^{13}$C-$CH_4$ trend changes by 0.04 ‰, almost half the observed $\delta^{13}$C-$CH_4$ trend during this period. Therefore,

constraining the initial conditions throughout the atmosphere are a serious challenge if $\delta^{13}$C-$CH_4$ observations are to be used to estimate the recent changes in the methane budget.

These first order sensitivity indices demonstrate several key challenges in methane inverse modelling studies. Three parameters that the mole fraction and $\delta^{13}$C-$CH_4$ are highly sensitive to, are often not explored in methane modelling: the OH loss is often assumed to be known (e.g. Schaefer et al. (2016); Worden et al. (2017)), as is the Cl loss (e.g. Nisbet et al. (2016); Rigby

et al. (2017)) or the Cl loss is left out completely (e.g. Turner et al. (2017)); and this work is the first, as far as we are aware, to include freshwater emissions as an independent source. There has been increasing acknowledgement that OH and Cl could play important roles in methane modelling (e.g. Rigby et al. (2017); Turner et al. (2017); Thanwerdas et al. (2019); Strode et al. (2020)), but the role of freshwater methane emissions has not received the same level of attention. This lack of attention is presumably the result of the freshwater source's large uncertainty, but it is this large uncertainty that makes this source so im-

portant in constraining the methane budget. The first order sensitivity indices also demonstrate that the atmospheric $\delta^{13}$C-$CH_4$ is sensitive to some parameters to which the mole fraction is relatively insensitive, so should provide additional complementary information. However, $\delta^{13}$C-$CH_4$ is also highly sensitive to the initial conditions and some source signatures (e.g. agriculture), which need to be accounted for to realise the value for global scale studies using these isotopic measurements. Furthermore, these sources of uncertainty need to be carefully considered in methane modelling studies that use $\delta^{13}$C-$CH_4$, because erro-

neous assumptions of well known initial conditions, source $\delta^{13}$C-$CH_4$, or kinetic isotope effects could have substantial impacts on top-down budget estimates.

### 3.5.2 Parameter interactions

The interaction between parameters is calculated by subtracting the first order sensitivity (Eq. 9) from the total effect of each parameter (Eq. 10). The interaction of one particular parameter with all other parameters is the proportion of the output

variance explained by changing that parameter alongside all other parameters, removing the proportion of the output variance from changing that parameter independently of all other parameters. An example of interacting parameters is the OH loss and any source for the global mean mole fraction: a lower OH concentration causes a greater mole fraction increase from an increase in emissions.

The parameter interactions are shown in Fig. 8. These interactions are generally small, with the largest being 3 %. The

interactions across all parameters account for 9 % of the output variance in the $\delta^{13}$C-$CH_4$ inter-hemispheric difference, and at most 1 % for the other five outputs. This means that we can essentially consider the effect of each parameter independently in





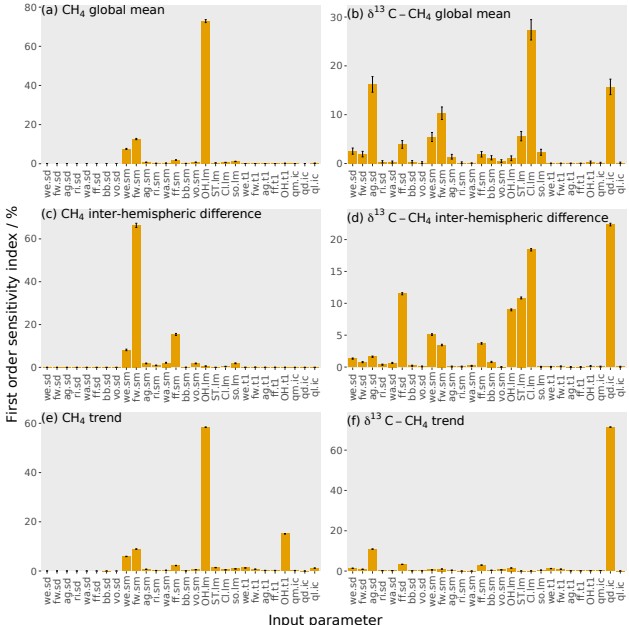

**Figure 7.** The orange bars show the first order sensitivity coefficients to the input parameters, with the error bars showing the uncertainty in these indices (calculated using bootstrap resampling, see Supplement). Each panel is for one of six outputs: (a) mole fraction global mean, (b) $\delta^{13}$C-CH$_4$ global mean, (c) mole fraction inter-hemispheric difference, (d) $\delta^{13}$C-CH$_4$ inter-hemispheric difference, (e) mole fraction trend, and (f) $\delta^{13}$C-CH$_4$ trend. The values given here are for the temporal mean of the time series. The input parameter codes are given by a combination of a two character code giving the source or loss, (wetlands (we), fresh water (fw), agriculture (ag), rice (ri), waste (wa), fossil fuels (ff), biomass burning (bb), volcanoes (vo), hydroxyl radical (OH), stratospheric (ST), Cl radical (Cl), soil (so), total source magnitude (qm), total source $\delta^{13}$C-CH$_4$ (qd), total loss imbalance (ql)) and another code giving the type of parameter, (source $\delta^{13}$C-CH$_4$ (sd), source magnitude (sm), loss magnitude (lm), temporal trend (t1), initial condition (ic)).

this sensitivity analysis. For this complex simulator, one-at-a-time sensitivity tests would produce a similar result, though this will not necessarily be the case for other models (Saltelli and Annoni, 2010).

These interactions are small in terms of a sensitivity analysis looking for the parameters that cause the greatest proportion
of the output variance. For example, parameter interactions account for 0.2 % and 0.9 % of global mean mole fraction and $\delta^{13}$C-CH$_4$ output variance, respectively. However, the parameter interactions must be considered in order to build an accurate emulator. For example, the 0.2 % and 0.9 % output variance is equivalent to a standard deviation of 10 ppb and 0.13 ‰ in the output, which are large compared to the quantities that the emulator is trying to predict (e.g. inter-hemispheric difference or trends). These values account for most of the difference in performance of the Gaussian process and multiple linear regression,
which does not consider parameter interactions, in Sect. 3.4.





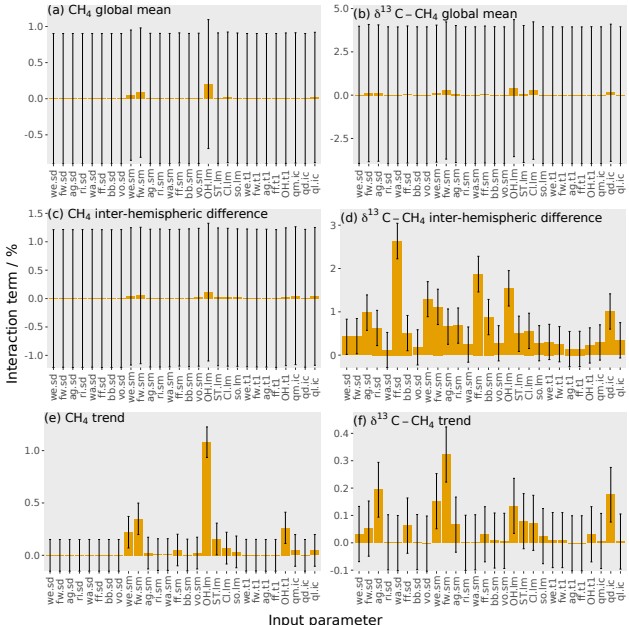

**Figure 8.** The orange bars show the interaction terms of the parameters with the error bars showing the uncertainty in these interactions (calculated using bootstrap resampling, see Supplement). Each panel shows one output: (a) mole fraction global mean, (b) $\delta^{13}$C-CH$_4$ global mean, (c) mole fraction inter-hemispheric difference, (d) $\delta^{13}$C-CH$_4$ inter-hemispheric difference, (e) mole fraction trend, and (f) $\delta^{13}$C-CH$_4$ trend. The values given here are for the temporal of the time series. The input parameter codes are given by a combination of a two character code giving the source or loss, (wetlands (we), fresh water (fw), agriculture (ag), rice (ri), waste (wa), fossil fuels (ff), biomass burning (bb), volcanoes (vo), hydroxyl radical (OH), stratospheric (ST), Cl radical (Cl), soil (so), total source magnitude (qm), total source $\delta^{13}$C-CH$_4$ (qd), total loss imbalance (ql)) and another code giving the type of parameter, (source $\delta^{13}$C-CH$_4$ (sd), source magnitude (sm), loss magnitude (lm), temporal trend (t1), initial condition (ic)).

## 4 Conclusions

We have shown that Gaussian processes allow emulation of a global 3D chemical transport model (CTM) of atmospheric methane, producing a fast and accurate approximation of the response of methane mole fraction and $\delta^{13}$C-CH$_4$ to changes in model input parameters. In this work, 28 parameters were investigated, related to methane sources and sinks, based on

270 forward model simulations. However, we found that, compared to an estimate of model uncertainty, an accurate emulator could be built for this system using fewer than 100 training runs. Our model uncertainty estimate, which we term "invariant parameter error" was based on an ensemble of model runs in which several minor sources and sinks were perturbed within their estimated uncertainty ranges, showing that they could, when considered together, lead to a substantial, and often ignored, source of uncertainty in global methane modelling studies (with mean uncertainties in hemispheric methane and $\delta^{13}$C-CH$_4$

between 2000 and 2012 of approximately 10 ppb and 0.1 ‰, respectively).



We show that a Gaussian process outperforms multiple linear regression in emulating the 3D CTM methane simulations: the Gaussian process RMSE is small (1.0 ppb, 0.06 ‰) compared to the invariant parameter error, whereas the multiple linear regression error (18 ppb, 0.17 ‰) is larger. Therefore, the use of Gaussian process emulators does not much reduce how precisely the model matches observations, but multiple linear regression could. The poor performance of the multiple linear regression is primarily because of the parameter interactions and the non-linearity in the response of the mole fraction to the OH loss.

The speed of emulation allows many more 3D CTM outputs to be generated than would be possible running the CTM itself, allowing a wider range of possible analyses. In this work, a thorough sensitivity analysis was carried out, which required millions of runs of the emulator. The sensitivity analysis demonstrated some of the issues with current methane modelling. The OH loss, Cl loss, and freshwater source are frequently held constant or not included in methane modelling studies, but the mole fraction or $\delta^{13}$C-CH$_4$ outputs are highly sensitive to these parameters. Our analysis shows that $\delta^{13}$C-CH$_4$ measurements provide somewhat independent constraints on the sources and sinks of methane, as they are sensitive to different model parameters. However, several of these parameters are $\delta^{13}$C-CH$_4$-specific so do not provide information on the methane budget alone. In particular, $\delta^{13}$C-CH$_4$ is highly sensitive to its initial conditions, which must therefore be very well constrained so as not to bias modelled trends, even over almost two decades.

Whilst we have focused here on a variance based sensitivity analysis, we anticipate that there could be multiple future applications of an accurate and fast emulator of 3D CTM simulations of atmospheric methane. This system could allow for the calculation of input parameter values that are consistent with observations (history matching), or could allow us to determine the set of parameter values that optimally simulate observations (e.g. through Bayesian optimisation). While in this work hemispheric emulators were created, it is also possible to emulate individual grid cells in the 3D CTM, which would provide a more accurate representation of the 3D CTM output. This number of emulators is feasible as the same training dataset could be used, and the computational burden of both building and running the emulator is far smaller than creating the 3D CTM training simulations. This allows new and flexible emulators to be built, and used for novel applications, without the need to rerun the 3D CTM.

## 5 Code and data availability

The code used to create the freshwater emissions field and the field itself are available at https://doi.org/10.17605/OSF.IO/Q9F8P (Stell, 2020a). The code and datasets used to train the emulators and carry out the sensitivity analysis are available at https://doi.org/10.17605/OSF.IO/Z435M (Stell, 2020b).

*Author contributions.* All authors contributed to the conception and development of the project, A.S wrote the code and performed the calculations, and all authors contributed to the manuscript.





*Competing interests.* The authors declare that they have no conflict of interest.

*Acknowledgements.* A.S. was funded under a Natural Environment Research Council (NERC) studentship through the Great Western 4+ Doctoral Training Partnership. L.W. and M.R. were funded by the NERC Methane Observations and Yearly Assessments (MOYA) highlight topic (NE/N016548/1) and NERC grant NE/M014851/1. Model simulations were carried out using the University of Bristol BlueCrystal high-performance computing system, and analysis was carried out using hardware funded under NERC grant NE/L013088/1.




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
