# Peer review of "Atmospheric methane source and sink sensitivity analysis using Gaussian process emulation"

_Atmospheric Chemistry and Physics, 2020_

## Referee Comment (RC1) · Anonymous Referee #1 · 8 Sep 2020

This paper uses Gaussian process emulation to explore the sensitivity of simulated methane to the uncertainties in multiple parameters. The greater computational speed of the emulator compared to the 3-dimensional model it emulates allows a more thorough exploration of parameter space than would be possible with the original model. This is a state-of-the art method and the study brings some interesting insights to the long-standing challenge of constraining the methane budget, such as the importance of freshwater emissions. Consequently, it would be a useful addition to the literature. However, some clarifications and greater use of observations would strengthen the paper, as noted in the comments below.

[Figure]

General comments

1. While the focus of this study is understanding model sensitivities, it would be useful to include more comparisons to observations to demonstrate whether the model sensitivities are reasonably realistic. In other words, if the model shows high sensitivity to a particular source or sink, are we confident that methane observations are really that sensitive to that source or sink? This information is difficult to determine from Fig. 3. Perhaps showing the model has reasonable skill in capturing interannual variability at a site heavily influenced by biomass burning or by wetland emissions would help demonstrate a realistic level of sensitivity to those sources.

2. The large number of 3D model simulations used to train and test the emulator is itself a substantial effort and potentially a valuable resource. Could these simulations provide additional information to support the analysis? For example, this study focuses on just hemisphere or global averaged measures of methane, but the 3D model fields could potentially take greater advantage of geographic differences.

3. Section 2.5: Please justify why the uncertainty in the invariant parameters is a good estimate of the CTM error, and compare to the error you would get from the model-data mismatch.

4. Since the initial conditions for the isotopic composition are listed as one of the important quantities to constrain, more detail is needed regarding how the initial conditions are specified in the model simulations. Are observations used in any way to constrain the initial state?

Specific Comments

1. Line 30: Please rephrase without parentheses

2. Lines 58-62: Another reference relevant to this work is: Wild, O., Voulgarakis, A., O'Connor, F., Lamarque, J.-F., Ryan, E. M., and Lee, L.: Global sensitivity analysis of chemistry–climate model budgets of tropospheric ozone and OH: exploring model

diversity, Atmos. Chem. Phys., 20, 4047–4058, https://doi.org/10.5194/acp-20-4047-2020, 2020

3. Line 134: Does this spin-up lead to a reasonable reproduction of surface observations in the early portion of the time period?

4. How was the number of simulations chosen? It might help to refer to the Supplemental Figure S8 here.

5. Line 202: Please explain the difference between x and x*

6. Line 277: What is the meaning of "arbitrary initial condition range"?

7. Line 341: What is the "initial condition source del-13C"? Do you mean the initial conditions for the del13C values of atmospheric methane? Or are you talking about an emission source?

8. Lines 340-355: Isn't the initial condition at least partially constrained by surface observations?

9. Line 360: It would be nice to know the sensitivity to the assumption of hemispheric parity in OH

10. Line 373: Do you mean the magnitude of the agricultural source or its trend?

11. Line 414: Is this because the trends and hemispheric differences are themselves small compared to the mean?

12. Fig. S8: Why does the plot have only 3 points?

---

## Referee Comment (RC2) · Anonymous Referee #2 · 22 Sep 2020

This manuscript describes a method for testing the sensitivity of various atmospheric CH4 metrics to uncertainties in its budget components. The method generates Gaussian process emulators of hemispheric, monthly mean CH4 mole fraction and $\delta$13C-CH4, using as inputs the same parameters used to initialize a 3-D chemical transport model. Once trained on and validated against 3-D CTM output, emulators are compared against multiple linear regression and a measure of the CTM uncertainty. The emulators are then applied by conducting millions of simulations covering the full parameter space of the 28 inputs. The sensitivities of CH4 and $\delta$13C-CH4 to each input are quantified based on the emulator simulations.

[Figure]

Overall, this manuscript presents a novel, insightful approach to understanding uncertainties in the simulated atmospheric CH4 budget. The text is well organized and clearly written, references to the previous literature thorough, and presentation of figures clear. The topic is pertinent to ACP; after the authors have considered several comments, detailed below, this article should be suitable for publication.

Major Comments

My only major suggestion is that perhaps the observational dataset, currently just used to show that the CTM simulations encompass realistic values, could be incorporated into evaluation of the emulator simulations. For instance, looking at Fig. 4, the CTM appears to underestimate the observed global mean $\delta$13C-CH4 value considerably (panel b). Could the emulator simulations be used to posit the drivers of the CTM underestimate? I understand that it would be unreasonable to meaningfully look at millions of simulations one-by-one, but perhaps the optimal values of the largest drivers of global mean $\delta$13C-CH4 (from Fig. 7b) could be identified? I.e., which combinations of inputs are needed to close in on the observed global mean $\delta$13C-CH4? This could be done for all the observed metrics shown in Fig. 4, if sorting through the emulator simulations to find observation-matching values is feasible.

Minor Comments

L78: While Gaussian process emulation has not been used for study of the methane budget specifically (as far as I am aware), it was recently used to evaluate the CH4 lifetime due to loss by OH. Please see and cite Wild et al., Global sensitivity analysis of chemistry–climate model budgets of tropospheric ozone and OH: exploring model diversity, https://doi.org/10.5194/acp-20-4047-2020

L93: The authors allude here to the Gaussian process inputs being maintained in their original spatial resolution. Does this mean all inputs are 2-D fields at 12x11.25 degrees resolution? Or are some 3-D? An explicit statement of exactly what is being fed into the Gaussian process emulators would be helpful, particularly regarding the inputs'

dimensionality.

Table 2: For the Trend values in the final column, the units are given as "%". Since trends are usually expressed as a rate, I would recommend noting the time period (I believe 2000-2012, based on my interpretation of the text) in the Table header information.

L195: I would hesitate to say that the loss of CH4 by OH is linear; the abundance/loss of CH4 has a feedback on the abundance of OH (see, e.g., Holmes et al., JAMES, https://doi.org/10.1002/2017MS001196). This would likely only influence results regarding large perturbations to CH4, so may not be relevant here, but it should probably be noted.

L360: It would be interesting to assess the role of altered spatial distributions of OH, both in the horizontal (i.e., more NH OH as many global models simulate) and in the vertical (i.e., what if there's more OH in the free troposphere than anticipated by Spivakovsky et al.?). It is understandable if this is beyond the scope of the current study but would make a good future direction.

L380: "are a serious" should be "is a serious"

L388: I would be interested to see a bit more discussion regarding the freshwater source of CH4. Some context regarding what is known about these emissions (that these are distinct from wetlands, what we know about the mechanism (bacteria?), that they are perhaps close in magnitude to wetlands emissions, etc.) would be helpful to the reader without them having to refer back to Saunois et al. This is potentially a very interesting finding, and some context could help raise awareness of this issue in the community.

NB: both "fresh water" and "freshwater" are used in several locations; I suggest maintaining consistency.

---

## Author Comment (AC1) · 27 Nov 2020

We would like to thank the reviewer for their helpful comments. In this document, we reply to each comment, providing extra detail and outlining how we have updated the manuscript.

The main suggestion from both reviewers was to make greater use of observations. This is a valid point, and we thought about this a lot before submitting this paper. We decided against going down this route because we felt that the most effective way to combine model sensitivities (in this case derived using Gaussian process emulation) with observations is through a full Bayesian inverse analysis. This will require some

additional methodological development (to effectively make use of the Gaussian process) and much more involved consideration of model and prior uncertainties. We felt that adding this material would make the paper long, less readable, and may take focus away from the emulation method and the sensitivity analysis, which we feel are novel and important in their own right. Therefore, we hope the reviewers will agree with our suggestion that a full inverse analysis would best be presented in a follow-up paper, which is currently in preparation.

***While the focus of this study is understanding model sensitivities, it would be useful to include more comparisons to observations to demonstrate whether the model sensitivities are reasonably realistic. In other words, if the model shows high sensitivity to a particular source or sink, are we confident that methane observations are really that sensitive to that source or sink? This information is difficult to determine from Fig. 3. Perhaps showing the model has reasonable skill in capturing interannual variability at a site heavily influenced by biomass burning or by wetland emissions would help demonstrate a realistic level of sensitivity to those sources.***

Following on from our comment regarding the use of observations above, we argue that a detailed comparison with observations would not change the main outcomes of this paper. The MOZART model, run in a very similar configuration has already been extensively compared to other models and observations in previous work (e.g. Patra et al., 2011, doi:10.5194/acp-11-12813-2011). Here, the main focus is to present a method for exploring model input-output relationships and estimating the relative importance of the uncertainty in the sources and sinks in driving hemispheric trends. The part of the paper concerning method development does not rely on model accuracy, and the hemispheric sensitivity analysis will only be weakly influenced by errors in site-specific mole fractions. On the latter point, we also note that we chose to include only model grid cells that contain background NOAA measurement stations. As such, none

of the model data points which were used to estimate hemispheric mole fractions are strongly influenced by nearby sources such as biomass burning or wetland emissions.

Again, we propose that a more detailed site-specific measurement comparison would be better placed in a follow-up paper where an inverse analysis was performed to constrain the model using atmospheric observations.

To make the point that a detailed comparison to observations has been performed previously, to line 117 we have added: "The MOZART model, run in a similar configuration, has been used previously in global methane studies and has been compared to other models and to observations (e.g. Patra et al., 2011)."

*The large number of 3D model simulations used to train and test the emulator is itself a substantial effort and potentially a valuable resource. Could these simulations provide additional information to support the analysis? For example, this study focuses on just hemisphere or global averaged measures of methane, but the 3D model fields could potentially take greater advantage of geographic differences.*

We agree, there will be further information contained in site-to-site differences that we have not explored here. The reason we did not introduce additional metrics, further than absolute global mean mole fraction or $\delta^{13}$C-CH$_4$, and their trends and inter-hemispheric gradients, was simply because it would have made the sensitivity discussion more difficult to understand, but with diminishing returns (i.e. the finer scale you go, the subtler the sensitivity information becomes). As mentioned in the paper, using our method, it is trivial to emulate individual grid cells of the model in order to fully utilise the 3D nature of the simulations. This is something that we aim to investigate further in our future inversion paper.

We acknowledge that there may be useful features in our model ensemble that other

researchers wish to explore, and for that reason, we have made the processed training dataset available via OSF (https://doi.org/10.17605/OSF.IO/Z435M). We would also be happy to share the raw MOZART output if requested. This raw output has not been shared publicly as it is hundreds of gigabytes of data, and reproducible following the steps in the paper.

***Section 2.5: Please justify why the uncertainty in the invariant parameters is a good estimate of the CTM error, and compare to the error you would get from the model-data mismatch.***

We agree that this uncertainty comparison has been left out and is useful to include. Line 292 (now line 309 in the revised paper), reads: "These values are slightly larger than the estimate of the combined measurement and model representation uncertainty, which examines the limited temporal and spatial resolution of the model (further details in the Supplement). Additionally, the invariant parameter uncertainty is large compared to atmospheric methane trends (e.g. between 2000 and 2012, the methane mole fraction and $\delta^{13}$C-CH$_4$ changed by around 40 ppb and -0.1 ‰ respectively)."

The calculation of the measurement and model representation uncertainty is additionally added to the Supplement, in a new section called 'The measurement and model representation uncertainty':

The calculation of the combined measurement and model representation uncertainty, from now on referred to as the model-measurement discrepancy uncertainty for brevity, is detailed here. The model-measurement discrepancy uncertainty is calculated by considering four elements that would cause the model output to differ from the observations: the measurement uncertainty, the model representation uncertainty, the different stations sampling in each month of the observations, and the different sample times in the observations. To account for these differences, the standard deviation in 10 000 samples from the uncertainty distributions of these four elements is calculated.

The two MOZART simulations in the training dataset closest to the methane mole fraction and $\delta^{13}$C-CH$_4$ observations were chosen as base simulations, around which these uncertainties are examined.

To account for the measurement uncertainty, a random value drawn from a normal distribution with a mean of zero and the median standard deviation from the NOAA and INSTAAR datasets (1.7 ppb and 0.051 ‰ for the methane mole fraction and $\delta^{13}$C-CH$_4$, respectively). This random value is added to every 6-hourly output value in each grid cell of the base simulations, in each of the 10 000 samples.

To calculate the horizontal representation uncertainty, a higher spatial resolution (1.89° N × 2.50° W) MOZART simulation with the mean emissions and losses in the training dataset was used. The range of outputs over the high-resolution grid cells within a low-resolution grid cell was calculated. The vertical representation uncertainty is calculated by taking the range of the output in each low-resolution grid cell and the grid cell above and below. For each of the 10 000 samples, a random value drawn from a uniform distribution between minus half the range and plus half the range is added to every 6-hourly output value in each grid cell of the base simulations for both the horizontal and vertical representation uncertainty.

The model hemispheric time series includes all grid cells with measurement stations in every month, regardless of whether there are observations for that station in that month. Therefore, the effect of including different stations in the hemispheric mean is explored by bootstrap resampling. For each of the 10 000 samples, 25 stations for the methane mole fraction and 10 stations for the $\delta^{13}$C-CH$_4$ (the number of stations included in this study) were chosen by sampling the stations with replacement.

The model hemispheric monthly time series includes all 6-hourly outputs at a station, but the observation hemispheric time series includes only approximately four samples in a monthly mean. To include the effect of having differently timed samples in the monthly output, four random time points are chosen to contribute to each station's

monthly value in each of the 10 000 samples.

The hemispheric time series is then calculated, and the standard deviation in the 10 000 samples of the hemispheric time series is used as the model-measurement discrepancy uncertainty. This uncertainty has a median value of 5 ppb and 0.05 ‰ in the southern hemisphere, and 10 ppb and 0.08 ‰ in the northern hemisphere."

We did not intend to suggest that this invariant parameter uncertainty was a better estimate than those presented in previous studies. Instead, we propose that this is a type of uncertainty that can be readily quantified by the type of model ensemble we have used here, but which has, to the best of our knowledge, not been presented previously. The "true" model uncertainty will have components related to the invariant parameter uncertainties, representation errors and systematic model transport model errors, as stated in lines 240-244 (now lines 257-260 in the revised paper):

"This invariant parameter error does not include many other sources of error (e.g. model transport uncertainties are not addressed), and higher-order "invariant parameter errors" (e.g. erroneous trends or spatial distributions), so can be considered a lower bound of the total 3D CTM error."

***Since the initial conditions for the isotopic composition are listed as one of the important quantities to constrain, more detail is needed regarding how the initial conditions are specified in the model simulations. Are observations used in any way to constrain the initial state?***

Following the reviewer's helpful comments, we have decided to reduce the spin-up parameter range so that the $\delta^{13}$C-CH$_4$ sensitivity is not so strongly dominated by the spin-up parameters. This has been done by roughly matching the 1996 initial conditions to observations.

We also agree that the initial conditions should be better described, and further detail

has been added in the revised version of the paper. The paragraph beginning line 160 (now line 166) has been amended to:

"Three parameters were varied during the spin-up: the total source magnitude, the total source $\delta^{13}$C-CH$_4$ signature, and an overall imbalance between the source and sink. Table 2 gives the range of these spin-up parameters. The range of the spin-up total source magnitude was derived by considering the minimum and maximum of the sum of the sources in Table 2. The range of the total source $\delta^{13}$C-CH$_4$ signature is constrained to values where the resulting January 1996 initial condition field has a global surface $\delta^{13}$C-CH$_4$ approximately matching observations (-47.3$\pm$0.6 ‰. Similarly, the range of the imbalance between the source and sink is constrained to values where the resulting January 1996 initial condition field has a global surface methane mole fraction approximately matching observations (1760$\pm$30 ppb). However, the January 1996 initial condition can go beyond these observed ranges by varying the other two spin-up parameters. The range of initial condition values is larger than that considered in previous methane modelling studies and it therefore may be an overestimate. However, given that constraints are only typically provided based on surface observations, whereas the initial model fields are 3D, extending from the surface to the upper stratosphere, it is very difficult to determine how uncertain the initial conditions truly are."

We also noticed a discrepancy in our stated parameter ranges which has been corrected in the revised paper. Lines 146-147 (now lines 148-151) have been replaced by:

"The ranges of possible source magnitudes were based on the ranges of compiled literature values in Saunois et al., 2016. The minimum and maximum values from Saunois et al., 2016 have been decreased and increased, respectively, by 10 % in this work as Saunois et al., 2016 does not include the uncertainties in the compiled studies or outliers in their ranges. The ranges of possible $\delta^{13}$C-CH$_4$ source signatures were

the three standard deviation ranges in Schwietzke et al., 2016."

*Line 30: Please rephrase without parentheses*

The parenthesis has been removed in the revised version of this paper.

*Lines 58-62: Another reference relevant to this work is: Wild, O., Voulgar-akis, A., O'Connor, F., Lamarque, J.-F., Ryan, E. M., and Lee, L.: Global sensitivity analysis of chemistry–climate model budgets of tropospheric ozone and OH: exploring model diversity, Atmos. Chem. Phys., 20, 4047–4058, https://doi.org/10.5194/acp-20-4047-2020, 2020*

We thank the reviewer for reminding us of this important and relevant paper. This has been added to our introductory section.

*Line 134: Does this spin-up lead to a reasonable reproduction of surface observations in the early portion of the time period?*

We have constrained the spin-up parameters to more closely reproduce surface observations as described above. However, as stated above, this range is still large, but we also do not know what the "true" initial condition uncertainty is, given the need to specify a 3D delta-value field, which is informed only be surface observations. In any case, as we note in line 379 (now line 409): "A wide range of $\delta^{13}$C-CH$_4$ initial condition values (Table 2) are examined in this work, however the importance of the initial conditions applies to even small ranges. For example, if the $\delta^{13}$C-CH$_4$ initial condition is perturbed by 0.1 ‰ from the initial median parameter values, the output atmospheric $\delta^{13}$C-CH$_4$ trend changes by 0.04 ‰ almost half the observed $\delta^{13}$C-CH$_4$ trend during this period."

Further, we note here that a future inverse modelling study should allow us to better

constrain this term, by allowing the early measurements to inform the initial conditions.

***How was the number of simulations chosen? It might help to refer to the Supplemental Figure S8 here.***

Within the Supplement, we have added an introductory sentence to Sect. 5 to acknowledge a rule of thumb:

"As a rule of thumb, ten times the number of parameters is a good number of training simulations to train a Gaussian process (e.g. Loeppky et al., 2009). However, this is dependent on the model being emulated and hence the accuracy of emulators trained with different numbers of simulations is tested here."

The following has been added after line 170 (now line 184) in the revised version of the paper:

"We chose 270 simulations as it was found to provide a balance between the accuracy of the emulator and the computational expense of generating training simulations. This is further discussed in the Supplement."

***Line 202: Please explain the difference between $\vec{x}$ and $\vec{x}*$***

$\vec{x}*$ are the input parameters to be predicted and $\vec{x}$ are the input parameters of the training dataset. This is already stated in lines 202 and 205 (now lines 219 and 222). Please let us know if we have misunderstood the confusion here.

***Line 277: What is the meaning of "arbitrary initial condition range"?***

We hope that our response to the reviewer's fourth point adequately explains this. In the revised version of the paper we have added a reference back to this new paragraph. Additionally, "arbitrary" was replaced by "large" for clarity.

**Line 341: What is the "initial condition source del-13C"? Do you mean the initial conditions for the del13C values of atmospheric methane? Or are you talking about an emission source?**

This, and throughout, has been changed to "spin-up source $\delta^{13}$C-CH$_4$ signature" for consistency with Table 2 and better clarity.

**Lines 340-355: Isn't the initial condition at least partially constrained by surface observations?**

This has been clarified in our response to the reviewer's fourth point.

**Line 360: It would be nice to know the sensitivity to the assumption of hemispheric parity in OH**

We agree that this is a potentially important factor that is not accounted for in this work. It was omitted in our emulator design, as we made the decision early on not to include spatial source or sink variations, focusing instead on magnitudes and temporal trends. However, some might argue that it should have been included, perhaps along with modifications to some source distributions, and potentially at the expense of some other terms. We accept this is a limitation of this work and it is discussed in lines 280-285 (now lines 297-302):

"Ideally, the spatial distributions of the emissions and losses would also be parameterised, allowing greater variation in the inter-hemispheric differences. However, only a limited number of parameters can be included in the Gaussian process emulation method of this work. The more parameters, the more 3D CTM simulations are required to train the emulator and the slower computation becomes. Therefore, only up to about 30 parameters are typically included in a Gaussian process, whereas methods such as adjoint models (e.g. Bousquet et al. (2011); Bergamaschi et al. (2013)) can include thousands of parameters."

We have also added the following sentence to line 361 (now line 385) to clarify this:

"However, had the uncertainty in the hemispheric distribution of OH been included in our analysis, it would likely have explained a larger proportion of this sensitivity."

**Line 373: Do you mean the magnitude of the agricultural source or its trend?**

We have clarified this to say "the agricultural source $\delta^{13}$C-CH$_4$ signature".

**Line 414: Is this because the trends and hemispheric differences are themselves small compared to the mean?**

This sentence has been rewritten in the revised paper:

"Whilst these interactions are relatively unimportant in this sensitivity analysis, they must be considered in order to build an accurate emulator. For example, the 0.2 % and 0.7 % of the output variance explained by parameter interactions for the global mean mole fraction and $\delta^{13}$C-CH$_4$, respectively, is equivalent to a standard deviation of 10 ppb and 0.09 ‰ in the output. This accounts for most of the difference in performance of the Gaussian process and multiple linear regression, which does not consider parameter interactions, in Sect. 3.4."

**Fig. S8: Why does the plot have only 3 points?**

This should have been clarified in the paper and the following text has been added to the figure caption:

"There are only three points as each point requires a new Latin hypercube design in order to properly sample the parameter space with a different number of simulations (i.e. an arbitrary sub-set of the largest ensemble cannot be used for this purpose, as it would not be a true Latin hypercube). This means that each point requires a new set

of MOZART training simulations, which is computationally expensive to repeat multiple times. However, this function is very unlikely to have multiple minima, and so we think this figure is enough to act as a rough guide."

---

## Author Comment (AC2) · 27 Nov 2020

We would like to thank the reviewer for their helpful comments. In this document, we reply to each comment, providing extra detail and outlining how we have updated the manuscript.

*My only major suggestion is that perhaps the observational dataset, currently just used to show that the CTM simulations encompass realistic values, could be incorporated into evaluation of the emulator simulations. For instance, looking at Fig. 4, the CTM appears to underestimate the observed global mean $\delta^{13}$C-CH$_4$*

[Figure]

*value considerably (panel b). Could the emulator simulations be used to posit the drivers of the CTM underestimate? I understand that it would be unreasonable to meaningfully look at millions of simulations one-by-one, but perhaps the optimal values of the largest drivers of global mean $\delta^{13}$C-CH$_4$ (from Fig. 7b) could be identified? I.e., which combinations of inputs are needed to close in on the observed global mean $\delta^{13}$C-CH$_4$? This could be done for all the observed metrics shown in Fig. 4, if sorting through the emulator simulations to find observation-matching values is feasible.*

This is a very similar point to the major comment by Reviewer 1, and one to which we gave a great deal of consideration before submitting the manuscript. As we wrote in our response to Reviewer 1:

We decided against going down this route because we felt that the most effective way to combine model sensitivities (in this case derived using Gaussian process emulation) with observations is through a full Bayesian inverse analysis. This will require some additional methodological development (to effectively make use of the Gaussian process) and much more involved consideration of model and prior uncertainties. We felt that adding this material would make the paper long, less readable, and may take focus away from the emulation method and the sensitivity analysis, which we feel are novel and important in their own right. Therefore, we hope the reviewers will agree with our suggestion that a full inverse analysis would best be presented in a follow-up paper, which is currently in preparation.

To answer the more specific element of the reviewer's comment regarding the principal cause of disagreement with the observations, we note that the cause of the global mean $\delta^{13}$C-CH$_4$ offset can be considered qualitatively using the sensitivity analysis itself. The parameters that are responsible for the largest proportion of the output variance are the $\delta^{13}$C-CH$_4$ source signature of agriculture, the magnitude of the Cl loss, and the magnitude of the freshwater source. It is these parameters that the output is

most sensitive to that are most likely to be adjusted to reach the optimal solution, for example, in an inversion.

The reviewer also raises an interesting suggestion that the ensemble could be examined to find the subset that best agrees with the data. We have indeed tried such approach, e.g. "history matching" as referenced in line 443 (now line 476 in the revised paper), for example, by attempting to find some subset of the parameter space that is consistent, within some uncertainty, of the observations. However, we found that, given the high dimensionality, even with an efficient emulator, it was extremely expensive to derive a statistically meaningful ensemble from a purely random exploration of the space. Therefore, as we note in our response to Reviewer 1, we feel that the most promising approach will be a Bayesian method, which can be explored more thoroughly in a follow-up paper.

*While Gaussian process emulation has not been used for study of the methane budget specifically (as far as I am aware), it was recently used to evaluate the CH4 lifetime due to loss by OH. Please see and cite Wild et al., Global sensitivity analysis of chemistry–climate model budgets of tropospheric ozone and OH: exploring model diversity, https://doi.org/10.5194/acp-20-4047-2020*

We thank the reviewer for reminding of us of this important paper that should have been cited. We have added this reference to the revised version of the paper.

*L93: The authors allude here to the Gaussian process inputs being maintained in their original spatial resolution. Does this mean all inputs are 2-D fields at 12x11.25 degrees resolution? Or are some 3-D? An explicit statement of exactly what is being fed into the Gaussian process emulators would be helpful, particularly regarding the inputs' dimensionality.*

In terms of the input to MOZART, all input fields are interpolated to the model resolution

of 12.00° N × 11.25° W, with emissions (and the soil loss) being 2D, and the other losses being 3D.

In terms of the input to the Gaussian process, the inputs are scaling factors of these fields, i.e. the input to the Gaussian process is 28 numbers (one for each parameter), but this will respond as if the 12.00° N × 11.25° W field had been scaled.

This has been clarified in the revised paper by adding to line 129 (now line 131):

"The model input fields are 2D for sources and the soil sink, and 3D for the remaining sinks."

Additionally, the following has been added to line 201 (now line 218):

"In this work, the input parameters are the 28 scaling factors in Table 2, and the outputs are the MOZART hemispheric average mole fraction and $\delta^{13}$C-CH$_4$ values."

*Table 2: For the Trend values in the final column, the units are given as "%". Since trends are usually expressed as a rate, I would recommend noting the time period (I believe 2000-2012, based on my interpretation of the text) in the Table header information.*

We agree, the revised version of the paper has both the time period (1996-2012) and the units as % yr$^{-1}$ in Table 2.

*L195: I would hesitate to say that the loss of CH4 by OH is linear; the abundance/loss of CH4 has a feedback on the abundance of OH (see, e.g., Holmes et al., JAMES, https://doi.org/10.1002/2017MS001196). This would likely only influence results regarding large perturbations to CH4, so may not be relevant here, but it should probably be noted.*

We agree that the loss of CH4 by OH is non-linear, this is discussed in the paragraph starting line 322 (now line 341):

"The multiple linear regression accuracy can be improved by considering the non-linearity of the mole fraction with respect to the OH loss. By using a log-transformed OH parameter to estimate the mole fraction, the RMSE becomes 11 ppb (the complete residual distribution is shown in Fig. 6). Multiple linear regression using a log-transformed OH parameter still has a significantly larger RMSE than the Gaussian process, implying that the remaining small non-linearities and parameter interactions are important for predicting the output value. This finding suggests that inverse modelling studies that have assumed linear and independent sensitivities between observations and source and sink parameters may have under-estimated their posterior uncertainties."

The Gaussian process does not assume linearity, and the mean function in line 195 (now line 210) could equally be set to zero and it would perform similarly well. We have added the following to the revised paper to clarify this:

"A linear mean function does not stop the Gaussian process from being able to model non-linear relationships."

***L360: It would be interesting to assess the role of altered spatial distributions of OH, both in the horizontal (i.e., more NH OH as many global models simulate) and in the vertical (i.e., what if there's more OH in the free troposphere than anticipated by Spivakovsky et al.?). It is understandable if this is beyond the scope of the current study but would make a good future direction.***

We do agree that this would indeed be interesting. However, as we have noted in our response to Reviewer 1, it was not included in our emulator design, because we made the decision early on to focus on uncertain magnitudes and trends in sources and sinks, rather than spatial distributions. We have acknowledged as much in line

279 (now line 297) and have added another line to the revised manuscript when discussing the sensitivity of the interhemispheric difference to the input parameters (line 360, now line 385): "However, had the uncertainty in the hemispheric distribution of OH been included in our analysis, it would likely have explained a larger proportion of this sensitivity."

*L380: "are a serious" should be "is a serious"*

We agree, this has been changed in the revised version of the paper.

*L388: I would be interested to see a bit more discussion regarding the freshwater source of CH4. Some context regarding what is known about these emissions (that these are distinct from wetlands, what we know about the mechanism (bacteria?), that they are perhaps close in magnitude to wetlands emissions, etc.) would be helpful to the reader without them having to refer back to Saunois et al. This is potentially a very interesting finding, and some context could help raise awareness of this issue in the community.*

The following sentence has been added after line 388 (now line 419) to address this:

"Freshwater bodies emit methane by bacteria breaking down organic matter in an anaerobic environment, as in wetlands, and the freshwater emissions are potentially of similar magnitude to wetlands, but more uncertain (as seen in Fig. 1)."

*NB: both "fresh water" and "freshwater" are used in several locations; I suggest maintaining consistency.*

This is intentional as "fresh water" is a noun whereas "freshwater" is an adjective.